# High-Throughput Screening Identified Compounds Sensitizing Tumor Cells to Glucose Starvation in Culture and VEGF Inhibitors In Vivo

**DOI:** 10.3390/cancers11020156

**Published:** 2019-01-30

**Authors:** Ran Marciano, Manu Prasad, Tal Ievy, Sapir Tzadok, Gabriel Leprivier, Moshe Elkabets, Barak Rotblat

**Affiliations:** 1Department of Life Sciences, Ben-Gurion University of the Negev, Beer Sheva 84105, Israel; ranmarci@post.bgu.ac.il (R.M.); levyag@post.bgu.ac.il (T.I.); 2The National Institute for Biotechnology in the Negev, Beer Sheva 84105, Israel; 3The Shraga Segal Department of Microbiology, Immunology and Genetics, Faculty of Health Sciences, Ben-Gurion University of the Negev, Beer-Sheva 84105, Israel; mpmnedumudy@gmail.com (M.P.); sapirtza@post.bgu.ac.il (S.T.); 4Department of Pediatric Oncology, Hematology, and Clinical Immunology, Medical Faculty, University Hospital Düsseldorf, 40225 Düsseldorf, Germany; gabriel.leprivier@med.uni-duesseldorf.de

**Keywords:** cancer metabolism, drug combination, synthetic lethality, HTS, small molecule

## Abstract

Tumor cells utilize glucose to fuel their anabolic needs, including rapid proliferation. However, due to defective vasculature and increased glucose uptake, tumor cells must overcome glucose deprivation. Accordingly, tumor cells depend on cellular pathways promoting survival under such conditions. Targeting these survival mechanisms can thus serve as a new therapeutic strategy in oncology. As such, we sought to identify small-molecule inhibitors which sensitize tumor cells to glucose starvation by high-throughput drug screening in vitro. Specifically, we searched for inhibitors that selectively killed tumor cells growing in glucose-free but not in normal medium. This phenotypic drug screen of 7000 agents with MCF7 cells led to the identification of 67 potential candidates, 31 of which were validated individually. Among the identified compounds, we found a high number of compounds known to target mitochondria. The efficacies of two of the identified compounds, QNZ (EVP4593) and papaverine, were validated in four different tumor cell lines. We found that these agents inhibited the mTOR(Mechamistic\Mammilian Target of Rapamycin) pathway in tumor cells growing under glucose starvation, but not under normal conditions. The results were validated and confirmed in vivo, with QNZ and papaverine exhibiting superior antitumor activity in a tumor xenograft model when combined with the VEGF inhibitor bevacizumab (avastin). Administering these drug combinations (i.e., avastin and papaverine, and avastin and QNZ) led to significant reductions in proliferation and mTOR activity of the aggressive DLD1 colon cell line in mice. Given our findings, we propose that compounds targeting metabolically challenged tumors, such as inhibitors of mitochondrial activity, be considered as a therapeutic strategy in cancer.

## 1. Introduction

Tumor cells upregulate glucose metabolism and oxidative phosphorylation (OXPHOS) to support their enhanced anabolic demands [1,2]. Apart from glycolysis, glucose is also utilized by the pentose phosphate pathway, the hexosamine pathway, and the one-carbon metabolism pathway to generate building blocks and reducing power for the cell [1]. Indeed, glucose uptake is upregulated by various oncogenes, such as MYC [3] and RAS [4], with such tumor-driven high glucose uptake being exploited for diagnosis using radiolabelled glucose analogues such as ^18^F-FDG and positron emission tomography. However, a number of tumors develop within a glucose-depleted tumor microenvironment [4,5,6,7,8], as measurements of glucose levels in tumor tissues indicate a 2–45-fold reduction in glucose in such tumors compared to normal surrounding tissues [9,10]. Hence, these tumor cells must evolve proper cellular mechanisms to adapt to low-glucose conditions. 

In an unbiased shRNA-based (short hairpin RNA) screen targeting metabolic proteins specifically aimed at identifying proteins that support the survival of tumor cells in low-glucose-containing medium, Birsoy et al. identified mitochondrial OXPHOS proteins, and in particular, complex I proteins, as being central for the metabolic adaptation of tumor cells to low-glucose conditions [9]. In addition, other studies have identified several protective cellular pathways supporting the survival of tumor cells growing in glucose deprivation, including the unfolded protein response (UPR) pathway [11], the mTOR pathway [12,13,14,15,16], and the NF-ĸB pathway [17]. 

These requirements support the premise that proteins and cellular pathways which support cell survival under glucose starvation represent potential drug targets, as their inhibition is expected to selectively kill glucose-depleted tumor cells, but not normal tissues that do not experience glucose deprivation [9,18]. 

Based on the observations described above, we performed a phenotypic synthetic lethality high-throughput (HTP) drug screen in the hope of finding new potential anticancer drugs. In doing so, we identified compounds which selectively kill tumor cells growing in glucose-free media. Strikingly, we found significant enrichment for mitochondrial poisons. We validated the activity of two such compounds, QNZ (EVP4593) and papaverine, in culture and demonstrated that a combination of the antiangiogenic agent bevacizumab with QNZ or papaverine had superior antitumor activity in vivo than shown by either agent alone. These findings support the concept that pharmacological targeting of glucose-deprived tumor cells in vitro has the power to uncover compounds that can act synergistically with antiangiogenic agents to inhibit tumor growth in vivo.

## 2. Materials and Methods

### 2.1. Cell Cultures 

DLD, MCF7, PANC1, H1299, and U87 cells were cultured in Dulbecco’s modified Eagle’s medium (DMEM) (Biological Industries, Kibbutz Beit-Haemek, Israel) containing antibiotic–antimycotic (Tivan Biotech, Kfar-Saba, Israel), 1 mM sodium pyruvate, and 10% fetal bovine serum (FBS) (Biological Industries). All cell cultures were incubated in 5% CO_2_ at 37 °C. For glucose starvation, the medium was replaced with glucose-free DMEM containing antibiotic–antimycotic and 10% FBS. 

### 2.2. Cell Viability Assays

For compound screening, cells were plated on 384-well plates containing one compound/vehicle per well with or without glucose in the medium. The plates were incubated in 5% CO_2_ at 37 °C for 48 h. Cell viability was measured using a CellTiter-Glo Luminescent Cell Viability Assay kit (Promega, Madison, WI, USA). For validation of compound screening and determination of the IC_50_ values, cells were plated in 24-well plates with glucose-depleted medium and the compound of interest. The plates were incubated in 5% CO_2_ at 37 °C for 48 h, followed by aspirating the medium and washing with PBS. Two hundred and fifty microliters of crystal violet staining solution (0.5% (w/v) crystal violet powder, 20% methanol) was added to the plates. After incubation for 10 min at room temperature, the staining solution was discarded, the plates were washed 5 times with distilled water and dried, and 250 µL of 10% acetic acid was added to each well. Absorbance was measured at 570 nm.

### 2.3. Cell Death Assays

Cell death was determined using flow cytometry. Cells were harvested at 48 h post-treatments and stained with annexin V/PI using an apoptosis detection kit (Miltenyi Biotec, Bergisch Gladbach, Germany). Cells were analyzed using a flow cytometer (Sysmex, Kobe, Japan) equipped with FCS Express software according to the manufacturer’s instructions. 

### 2.4. Cell Lysis and Sample Preparation

Cell lysis was performed on ice. Cells were washed one time with PBS and scraped with RIPA lysis buffer (150 mM NaCl; 50 mM Tris pH = 8.0; 1% Triton X-100; 0.5% sodium deoxycholate; 0.1% SDS). Samples were sonicated and centrifuged at 4 °C for 20 min. The supernatant was collected and stored at −20 °C. Before proceeding, protein amounts were quantified using a Pierce^TM^ BCA Protein Assay Kit (Thermo Scientific, Waltham, MA, USA). 

### 2.5. Gel Electrophoresis and Immunoblotting 

Protein lysates were mixed with 5× sample loading buffer (250 mM Tris–HCl pH 6.8; 10% SDS; 30% glycerol; 10 mM DTT; 0.05% (w/v) bromophenol blue), followed by 10 min at 96 °C and spin-down. Samples were then loaded onto SDS-PAGE gels and gels were transferred on a nitrocellulose membrane. Membranes were blocked with blocking solution (5% skim milk; Tris buffer saline (TBS)) for 1 h. Membranes were incubated with primary antibody solution for 1 h at room temperature or overnight at 4 °C, followed by washing with TBS. Membranes were incubated with secondary antibodies for 1 h at room temperature. Images were developed using the Western blotting Serius HRP substrate kit and an electrochemiluminescence imager (Thermo Scientific, Waltham, MA, USA) and processed using GIMP 2.8 software. 

### 2.6. Antibodies 

HSC70 antibody (sc-7298) was obtained from Santa Cruz. Antibodies for phospho-p70S6K (Thr389) (9206s), total p70S6K (2708s), total 4EBP1 (C-9644S), phospho-S6RP (Ser240/244) (2215S), phospho-4EBP1 (Ser65) (9451S), and anti-rabbit (C-7074S) and anti-mouse (C-7076S) phosphor-AKT (Ser473) (9271S) were obtained from Cell Signaling Technology (Danvers, MA, USA). 

### 2.7. Tumor Xenografts and Drug Treatments in Vivo 

For the xenograft experiments, 6-week-old NOD–SCID (NOD.CB17-Prkdc^scid^/NCrCrl) mice were injected subcutaneously in the flank with DLD1 cells (5 × 10^6^ cells per injection in PBS). Two independent experiments were performed and each experiment consisted of 20 mice. All mice developed tumors. When the tumor volume reached 150–200 mm^3^, they were randomized into four groups of five mice in each experiment. Animals were treated with vehicle (PBS/saline), papaverine (10 mg/kg), or QNZ (EVP4593) (1 mg/kg) daily via intraperitoneal injection, or bevacizumab (5 mg/kg) twice weekly in combination with papaverine (10 mg/kg) or QNZ (1 mg/kg). Tumors were measured with digital caliper twice a week, and tumor volumes were determined using the formula: length × width^2^ × π/6. At the end of the experiment, animals were sacrificed using CO_2_ inhalation and the tumors were harvested for investigation. Tumor volumes were normalized to initial volumes and presented as an averaged percentage of the initial volumes ±standard error of the mean (SEM).

Mice were maintained and treated according to the institutional guidelines of Ben-Gurion University of the Negev. Mice were housed in air-filtered laminar flow cabinets with a 12-h light/dark cycle and food and water ad libitum. Animal experiments were approved by the Ben Gurion University of the Negev animal care and use committee (license number: IL.80-12-2015).

### 2.8. Immunohistochemistry (IHC) and Analysis

Following mice sacrifice, tumors were fixed in 4% paraformaldehyde. Tissues were dehydrated using alcohol gradient and embedded in paraffin. Five-micrometer sections were taken, deparaffinized, and dehydrated using xylene and alcohol gradients, respectively. The slides were incubated in 10 mM citric acid buffer, pH 6.0 at 100 °C for 20 min for antigen retrieval. The endogenous peroxidase activity was blocked with H_2_O_2_ (0.3%). Sections were then blocked for 1 h at room temperature with blocking solution (PBS, 0.1% TWEEN, 5% BSA), followed by incubation with primary antibodies. Ki67 (275R-1, Sigma, St. Louis, MO, USA), pS6RP (4857S, Cell Signaling, Danvers, MA, USA), and CD31 (ab28364, Abcam, Cambridge, UK) antibodies were diluted in blocking solution and incubated overnight at 4 °C. The ABC kit (VECTASTAIN Cat. VE-PK-6200) was used for detection according to the manufacturer’s protocol. Sections were counterstained with hematoxylin, dehydrated, and mounted with mounting media (Micromount, Leica, Cat. 380-1730, Wetzlar, Germany). IHC slides were digitalized using the Pannoramic Scanner (3DHISTECH, Budapest, Hungary) and analyzed using QuantCenter (3DHISTECH) using a single threshold parameter for all images of a specific staining sample in each experiment.

### 2.9. Statistics

Statistical analyses were done using GraphPad Prism 7.03 software. All cellular experiments were repeated at least three times. A two-tailed Student’s unpaired *t* test was performed to compare control versus treated group. *p* values of 0.05 (*), 0.01 (**), and 0.001 (***) were considered statistically significant. For experiments with more than two groups, a one-way ANOVA was calculated using Turkey’s multiple comparison test. In vivo experiments were performed with indicated n values, and a one-way ANOVA test was performed to compare between groups. 

## 3. Results

### 3.1. High-Throughput Synthetic Lethality Drug Screening for Selectively Potent Compounds under Glucose Starvation

To identify compounds specifically targeting tumor cells under glucose starvation conditions, we employed a strategy whereby cells were directly seeded in glucose-free or normal medium in 384-well plates containing the library of compounds being tested. This allowed the cells to grow for a given amount of time, after which cell viability was measured. Viability was compared between cells growing either in glucose-free or glucose-proficient medium in parallel plates (the workflow scheme is shown in Figure 1A). Any compound significantly reducing viability under glucose starvation but not in normal medium was considered a positive hit (depicted in yellow in Figure 1A). 

To set up the high troughput screen (HTP), we initially tested several conditions to determine the optimal number of cells for plating and time of treatment before measuring viability (Figure 1B). Specifically, we subjected the breast tumor cell line MCF7 to a 48-h period of treatment based on the evidence that glucose deprivation did not affect MCF7 viability when plated at 2000 cells/well within the first 48 h. Viability, however, dropped 72 h post-treatment (Figure 1B). Because MCF7 cells are fully viable after 48 h under glucose-starved conditions, any significant reduction in viability induced by a given compound at this time point would be indicative of it being a potential positive hit (Figure 1A).

Using these conditions, we screened 7000 compounds (Selleckchem, Munich, Germany). Upon plotting cell viability in normal versus glucose-free medium, we identified 67 compounds (Appendix A) which significantly reduced viability in glucose-free but not in normal medium (Figure 1C). Therefore, the HTP strategy employed here allowed us to identify compounds specifically targeting glucose-deprived tumor cells.

### 3.2. Screen Validation and Identification of QNZ and Papaverine as Compounds with Selective Toxicity under Glucose Starvation

We next validated the 67 identified compounds in terms of selectivity for glucose-deprived conditions by determining the half-maximal inhibitory concentration (IC_50_) for MCF7 cells grown in normal or glucose-free medium. Such analysis revealed that 31 compounds were significantly more toxic under glucose starvation conditions, as compared with growth in normal medium (Figure 2A; Appendix A). Strikingly, we found that ~50% of the positive hits corresponded to known mitochondrial toxins (Figure 2B; Appendix A). Our findings thus reinforce previous reports showing that cells are dependent on optimal mitochondrial activity to survive glucose starvation [9,19,20]. 

We further confirmed these findings in three tumor cell lines using two selected drugs, QNZ and papaverine, both known to inhibit mitochondrial activity [21,22]. The three tumor cells lines, namely the DLD1 (colon cancer), MCF7 (breast cancer), and U87 (brain cancer) cell lines, were cultured in glucose-containing or glucose-free medium, and viability was measured after applying increasing concentrations of QNZ and papaverine. Using this approach, we determined the IC_50_ of QNZ or papaverine on each cell line grown in glucose-free and normal media, and found that the efficacy of both agents was dependent on glucose levels (Figure 2B). Indeed, the IC_50_ for QNZ was calculated to be ~3 nM under glucose-deprived conditions, but could not be calculated under normal conditions (Figure 2B). Similarly, papaverine exhibited an IC_50_ of ~3 μM in glucose-free medium and <10 μM under normal conditions (Figure 2B).

For further validation, we retested the compounds QNZ and papaverine using DLD1 and U87 cells under glucose starvation and normal conditions, as well as two additional cell lines, i.e., the H1299 (lung cancer) and PANC1 (pancreatic cancer) lines (Figure 3A). We found that in all cases, tumor cells treated with QNZ or papaverine exhibited a greater reduction in viability under glucose starvation than when grown in glucose-proficient medium. 

Given that reduction in cell viability can result from increased cell death, we assessed the impact of QNZ and papaverine on cell death. Accordingly, we treated DLD1, MCF7, and U87 cells with QNZ or papaverine in the presence or absence of glucose in the medium and measured rates of cell death 48 h post-treatment using annexin V/PI staining and flow cytometry (Figure 3B). As expected, we detected extensive staining for annexin V in tumor cells treated with QNZ or papaverine upon glucose starvation, but not under normal conditions, indicating that these compounds specifically induced apoptosis under these restricted conditions. Together, these data indicate that QNZ and papaverine selectively kill tumor cells subjected to glucose deprivation. 

### 3.3. QNZ and Papaverine Inhibit the mTOR Pathway Selectively under Glucose Starvation

To obtain further molecular insight into the cellular response to QNZ and papaverine under glucose-starved conditions, we analyzed the activity of the mTOR pathway, which is a master regulator of the response to glucose starvation [12,13,14,15,16]. For this, DLD1, MCF7, and U87 cells treated with or without QNZ or papaverine in normal or glucose-free medium were analyzed for levels of phosphorylated-4EBP1, phosphorylated-P70S6K, and phosphorylated-S6RP, all reflective of mTOR pathway activity [23,24]. While a 3-h period of glucose starvation did not reduce phospho-4EBP1, phospho-P70S6K, or phospho-S6RP levels, likely due to the short period of the glucose challenge, we found that treatment with QNZ or papaverine led to a striking reduction in phosphorylated protein levels under these conditions, indicating inhibition of the mTOR pathway (Figure 4A). In contrast, treatment with any of the compounds had no effect on the levels of phospho-4EBP1, phospho-P70S6K, and phospho-S6RP under normal conditions (Figure 4A). These data indicate that cells respond to the combination of either QNZ or papaverine with glucose starvation by inhibiting the mTOR pathway. Furthermore, the phosphorylation of AKT was variable between cell lines, glucose levels, and the compounds (Figure 4A), supporting the notion that the mTOR pathway is specifically affected by QNZ and papaverine under glucose starvation.

We next explored if treatment of tumor cells with QNZ or papaverine affects mitochondrial activity under normal and glucose-starved conditions. Tetramethylrhodamine, ethyl ester (TMRE) and flow cytometry analysis showed that both QNZ and papaverine inhibited mitochondrial membrane potential regardless of whether the cells were glucose-starved or not (Figure 4B). The mitochondrial inhibitory activities of QNZ and papaverine are in accord with previous findings [21,22]. Because cells are dependent on mitochondrial activity for survival under glucose starvation, it is therefore likely that the selective toxicity of QNZ and papaverine towards glucose-starved cells is, at least in part, related to inhibition of mitochondrial activity. 

### 3.4. QNZ and Papaverine Enhance the Antitumor Activity of Bevacizumab in Vivo 

To evaluate the efficacy of QNZ and papaverine in targeting tumor development in vivo, we hypothesized that such compounds, which exhibit selective toxicity upon glucose starvation, would enhance the efficacy of drugs promoting metabolic stress in tumors. One such drug is the antiangiogenic bevacizumab (also known as avastin). To test our hypothesis, we established tumor xenografts in NOD–SCID mice using the aggressive DLD1 colon cancer cell line. When the tumors reached 100 mm^3^ in diameter, treatment was initiated. We followed tumor growth for 14 days, after which time the tumors were removed and their masses were measured. As shown in Figure 5A and B, the combination of QNZ or papaverine with bevacizumab, respectively, was significantly more potent than were the single agents, as reflecting by a slowing of tumor growth and the significantly less massive size of the tumors.

To further characterize the effect of our treatments on tumors, IHC analysis was performed. We found that the levels of the proliferation marker Ki-67 were lower in tumors treated with a combination of papaverine or QNZ and bevacizumab, as compared to the single agents alone. These results are in line with our findings that tumors subjected to combination treatments were smaller, as compared to those treated with papaverine or QNZ alone (Figure 5A,B). We also found that phosphorylated S6RP levels were lower in the tumors treated with the drug combination, as compared to treatment with single agents (Figure 5C,D), indicative of a reduction in mTOR pathway activity. These results are in line with our findings in cell culture, showing that when combined with glucose starvation, QNZ and papaverine inhibited the mTOR pathway (Figure 4A). Finally, we found significant reduction in the levels of CD31, a marker of endothelial cells, in all bevacizumab-treated tumors (Figure 5C,D), as would be expected from an antiangiogenic. This further supports our model, according to which, compounds with selective toxicity towards glucose-starved cells enhance the efficacy of drugs that induce metabolic stress in tumors in vivo.

## 4. Discussion

Tumor cells often grow within a metabolically deprived tumor microenvironment. This implies that tumor cells need to adapt to such conditions in order to develop into macroscopic tumors. In many cases, tumor cells exploit endogenous protective pathways for their own advantage, including those supporting survival under metabolic stress [14,23,25,26,27,28,29,30,31,32,33,34,35,36,37,38,39,40,41,42]. Tumor cells, however, become addicted to such survival mechanisms [6,11,17,43], thereby offering potential weaknesses that can be targeted. To uncover compounds targeting these adaptive mechanisms, we carried out a phenotypic drug screen in cells growing in glucose-deprivation versus normal conditions. This led to the identification of 67 compounds presenting selective toxicity towards glucose-starved tumor cells. We validated two of these drugs, papaverine and QNZ, which are both mitochondrial complex I inhibitors. Our data demonstrate that both papaverine and QNZ exhibited selective toxicity under glucose starvation, and more importantly, enhanced the efficacy of the antiangiogenic bevacizumab against tumor xenografts in vivo. Bevacizumab is an anti-VEGF inhibitor which inhibits the growth of blood vessels in colon cancer tumors [44] and is used clinically to treat colon cancer [44]. Our results thus highlight how compounds that sensitize tumors cells to glucose starvation can enhance the antitumorigenic activity of bevacizumab in vivo. This can lay the groundwork for further developing drug synergies between glucose deprivation-selective compounds and metabolic stress-inducing agents (such as antiangiogenics and 2-deoxyglucose) to improve tumor targeting.

The ability of tumor cells to survive glucose deprivation relies on profound metabolic reprogramming. This includes the blocking of anabolic processes, such as protein and fatty acid synthesis, and the activation of catabolic processes, such as fatty acid oxidation, together allowing cells to maintain redox balance and prevent energy depletion [1,33,45]. However, how mitochondrial activity responds to glucose-deprived conditions is still poorly understood. The induction of mitochondrial activity (i.e., oxygen consumption) in response to low-glucose conditions has been reported, but only in cells resistant to such growth conditions [9]. This suggests that an increase in mitochondrial activity may be linked to improved survival under glucose-deprived conditions. While glycolysis is less active under glucose starvation, it is possible that mitochondrial activity is enhanced to compensate and generate sufficient ATP and cofactors involved in redox reactions. Strikingly, a high proportion of the glucose-starved cell-selective compounds we identified are mitochondrial toxins (approximately 50% of the compounds). This supports the notion that tumor cells rely heavily on mitochondrial activity to survive and adapt to glucose deprivation. Therefore, employing mitochondrial toxins to selectively target tumor cells experiencing metabolic stress may be a promising therapeutic approach. Notably, a number of mitochondrial toxins have been successfully used to restrict tumor growth in vivo, such as doxorubicin [9] and metformin [46], although not in combination with metabolic stress-inducing agents.

Papaverine is a non-narcotic opiate alkaloid commonly used in the clinic for treatment of spasms [47] and erectile dysfunction [48]. Recently, papaverine was found to sensitize tumors to radiation therapy by inhibiting mitochondrial complex I, thus reducing tumor cell respiration and promoting tumor oxygenation [49]. We identified papaverine in our HTP screen for compounds sensitizing cells to glucose starvation in culture. In vivo, papaverine significantly enhanced the activity of bevacizumab in reducing tumor growth, thus providing support to our model. Interestingly, since papaverine is regularly used in the clinic and as bevacizumab is used as a first-line treatment in colon cancer, it is conceivable that the combination of the two would be beneficial and quickly available in patients. In addition, QNZ was not only shown to act as a mitochondrial complex I inhibitor in vitro [21], but also to inhibit NF-ĸB in a pathway known to promote survival under glucose starvation [17]. It is, therefore, possible that NF-ĸB inhibition may contribute to the selective toxicity of QNZ under glucose starvation.

Finally, we found that both papaverine and QNZ selectively inhibited the mTOR pathway in glucose-starved cells, which was confirmed in vivo in combined treatment with bevacizumab. Since mTOR inhibition is required to prevent cell death in response to glucose deprivation [12,16,50,51], these results may be indicative of the energy-depleted status of the cell. Indeed, by inhibiting mitochondrial activity, papaverine and QNZ may deplete mitochondrial ATP levels under glucose deprivation. Given that mTOR is a hub whose activity depends on ATP levels, this scenario would be expected to lead to an inhibition of the mTOR pathway, in accordance with our observations.

### Limitations of Our Study

We identified compounds exhibiting selective toxicity upon glucose starvation in vitro and found that two of the identified compounds synergize with avastin in inhibiting tumor growth in vivo. The synergic effect we observed in vivo may be due to the induction of glucose starvation by avastin treatment, but it may also be caused by hypoxia triggered by this compound or by a combination of both. It also remains to be determined whether such synergic action is specific to avastin or will occur with other antiangiogenic compounds and calorie-restriction mimetics. In addition, given that glucose-starved cells are dependent on mitochondria to survive under glucose starvation and since the compounds we identified are known inhibitors of mitochondrial complex 1, it is likely that QNZ and papaverine enhance the antitumoral activity of avastin by targeting the mitochondria in vivo. However, potential other mechanisms cannot be excluded until this hypothesis is formally tested.

## 5. Conclusions

In conclusion, our HTP drug screen confirmed the mitochondrial dependence of glucose-starved tumor cells. Two of the identified compounds were validated in vivo and found to enhance the antitumor activity of the antiangiogenic compound bevacizumab, highlighting a new concept whereby drugs targeting the mitochondria may enhance the activity of avastin.

## Figures and Tables

**Figure 1 cancers-11-00156-f001:**
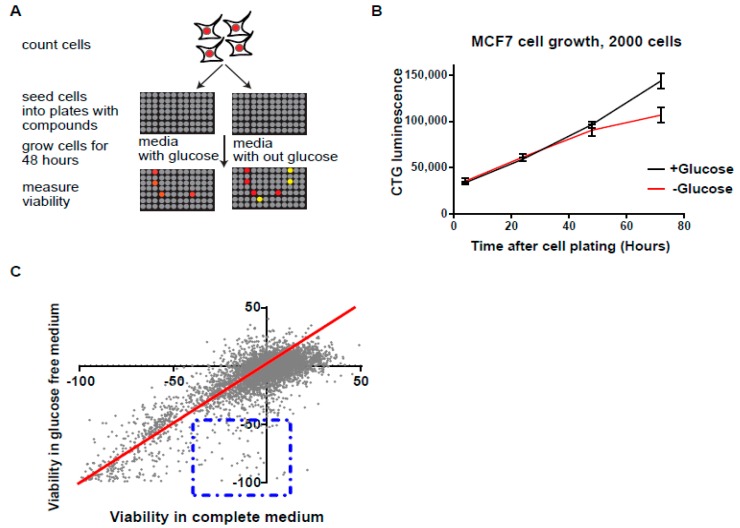
Highthroughput (HTP) screen for compounds selectively targeting cell viability under glucose starvation. (**A**) HTP drug screen pipeline. MCF7 cells were plated in glucose-free or normal medium in 384-well plates containing members of a compound library. Cell viability was measured after a given amount of time, and viability in glucose-free medium was compared with viability in normal medium. A positive hit was scored for compounds selectively reducing viability under glucose starvation. (**B**) Calibration of conditions used for HTP drug screening. At the indicated time points, the viability of MCF7 cells grown under glucose depletion was measured using a Cell-Titer-Glow kit (CTG). (**C**) HTP screening results. The viability of cells in normal and glucose-free media was plotted. Compounds exhibiting reduced viability under glucose starvation, as compared with normal conditions, are boxed.

**Figure 2 cancers-11-00156-f002:**
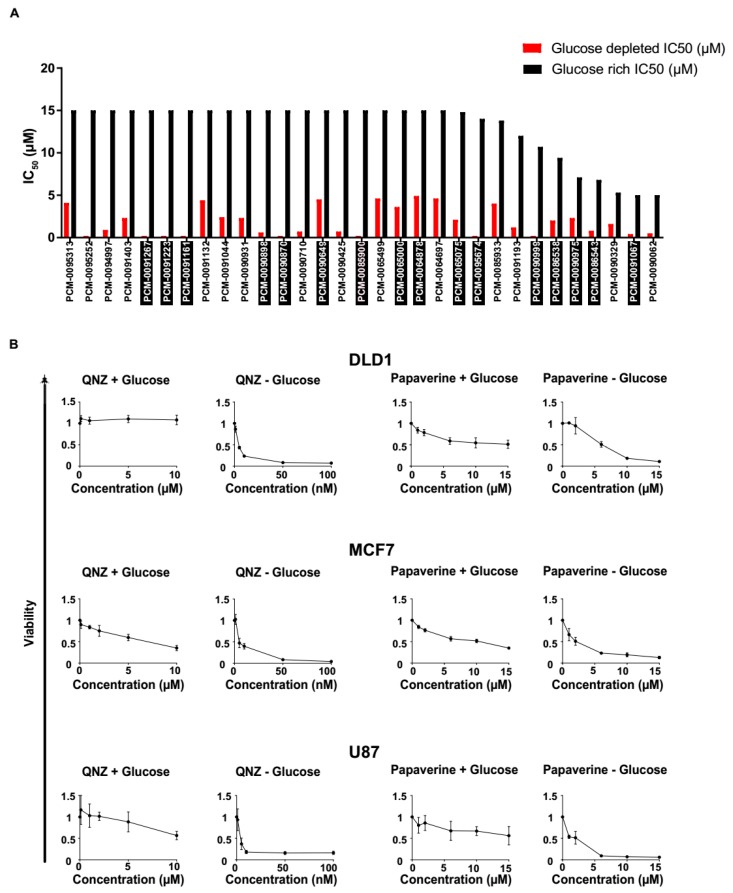
IC_50_ of the identified compound hits. (**A**) IC_50_ values of the 31 best hits were determined using a Cell-Titer-Glo kit. Compounds with black background are known to be mitochondrial poisons. (**B**) IC_50_ values of QNZ or papaverine with DLD1, MCF7 and U87 cells under the indicated conditions were determined using crystal violet staining. Data represents means ± SD; * *p* < 0.05; n ≥ 3 independent experiments.

**Figure 3 cancers-11-00156-f003:**
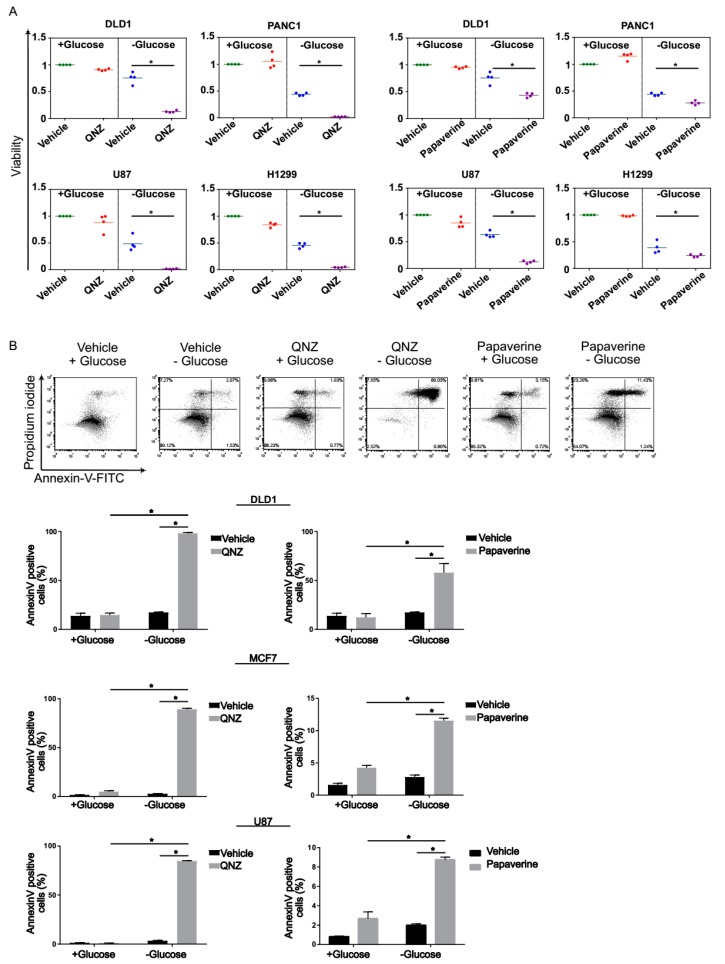
Selective killing of tumor cells under glucose starvation by QNZ and papaverine. (**A**) The viability of four tumor cell lines treated with QNZ (100 nM) or papaverine (3 µM) for 48 h under glucose starvation and normal conditions was determined using crystal violet staining. * *p* < 0.05. (**B**). Cell death of cells treated with the indicated compounds for 24 h in normal or glucose-starved medium was measured by annexin V/ Propidium iodide (PI) staining and fluorescence-activated cell sorting (FACS). Typical FACS plots obtained using MCF7 cells are shown. Data represent mean ± SD; * *p* < 0.05; n ≥ 3 independent experiments.

**Figure 4 cancers-11-00156-f004:**
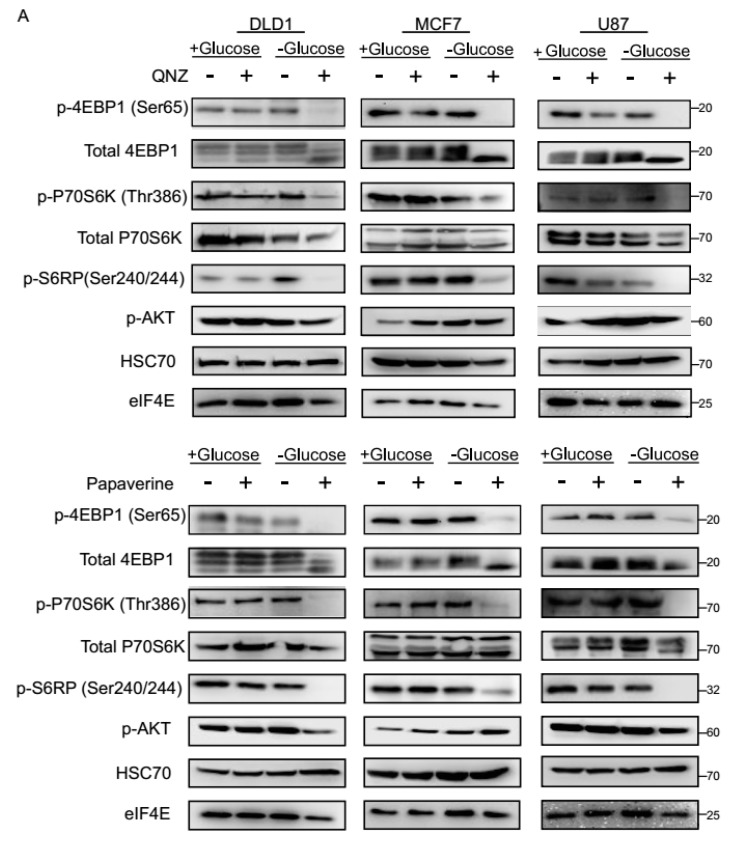
Acute inhibition of the mTOR pathway in tumor cells treated by papaverine and QNZ under glucose starvation. The indicated cells were treated for 3 h with papaverine (10 μM) or QNZ (100 nM) in the presence or absence of glucose, after which cell lysates were collected and analyzed by immunoblot using the indicated antibodies. (**B**) Mitochondrial activity of DLD1 cells treated as in (**A**) was measured by TMRE (Tetramethylrhodamine, ethyl ester) and FACS (fluorescence-activated cell sorting). AU = arbitarary units. Data represents mean ± SD; * *p* < 0.05; n ≥ 3 independent experiments.

**Figure 5 cancers-11-00156-f005:**
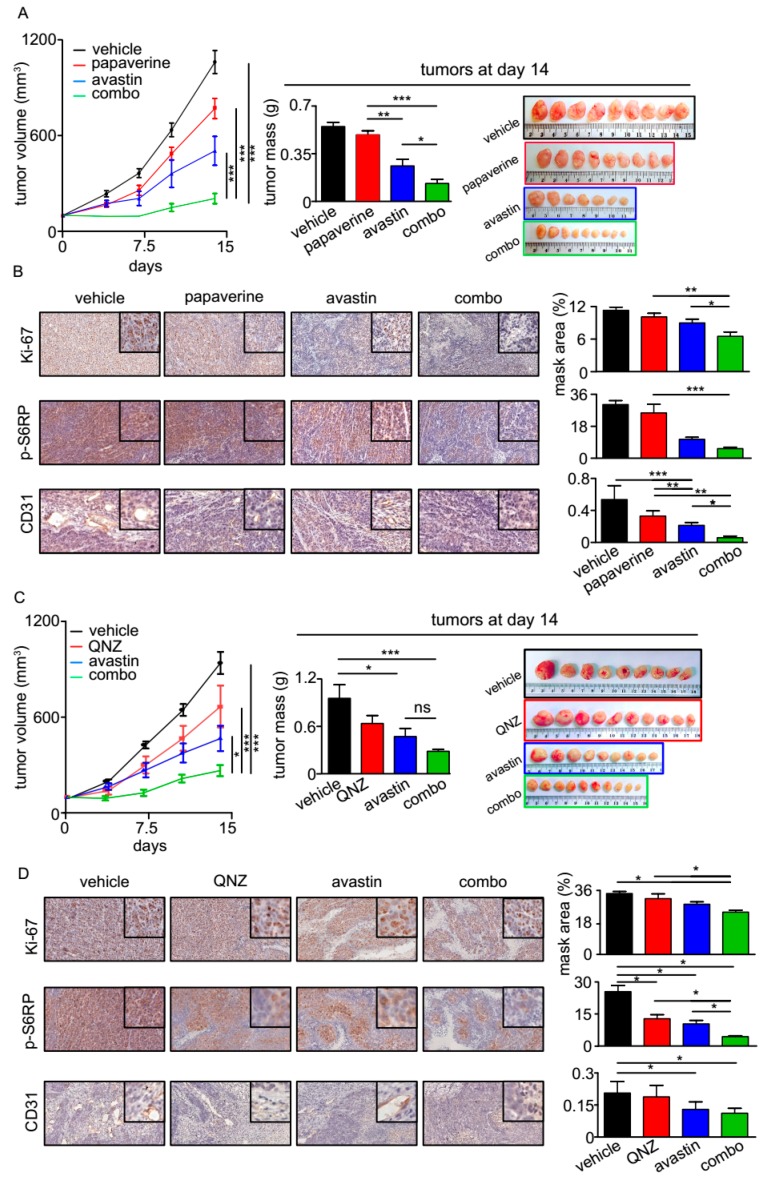
Combinations of bevacizumab with QNZ or papaverine delay tumor progression in vivo. (**A**,**C**) Tumor volumes of DLD1 cell xenografts in NOD–SCID mice; 5 × 10^6^ DLD1 tumor cells were injected subcutaneously. Mice were randomized into 4 groups (n = 9–10). Mice were treated with bevacizumab (5 mg/kg/2 w) via intraperitoneal (IP) injection along with either QNZ (left) (1 mg/kg/d) or papaverine (left) (10 mg/kg/d), administered by gavage. (**B**,**D**) Representative staining for Ki-67, p-S6RP, and CD31. Statistical significance was calculated on end points using one-way ANOVA.

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
