# Peer review of "High-Throughput Screening Identified Compounds Sensitizing Tumor Cells to Glucose Starvation in Culture and VEGF Inhibitors In Vivo"

_cancers, 2019, doi:10.3390/cancers11020156_

Round 1
Reviewer 1 Report
This manuscript by Marciano et al. describes an interesting new finding on the identification of compounds sensitizing tumor cells to glucose starvation in culture and to VEGF inhibitors in vivo. These findings are highly novel and original, and since VEGF inhibitors have until now limited success in the clinic, these new compounds could be of substantial interest as they may help to overcome treatment failure of VEGF inhibitors. However, there are a few concerns/questions that should be resolved before publication:
The authors should provide Western blot analysis for AKT in their cell line models with/without application inhibitors and glucose starvation. All Western blot micrographs should contain information on the molecular weight of the detected bands. Also, histological images should contain a scale bar.
The xenografts should be assessed for a differential induction of apoptosis (e.g. cleaved caspase 3) and/or necrosis (conventional H&E stains). No statistics are provided for differences in tumor mass in Figure 5c, as well as for CD31 stains in Figure 5d. These should be added.
Some figure labels and axes labels are very tine, especially in Figure 1. The authors should use the empty white space to enhance the readability of their figures by increasing the font size of the labels.
The discussion section should be further expanded also pointing out the limitations of the study.
References 30 and 35 are incorrectly spelled, and important reviews on 4EBP1 are not cited.
While the paper generally reads well, it contains numerous grammatical and orthographical errors (e.g. xerographs instead of xenografts; or incorrect use of verbs and nouns in divergent plural/singular form). Hence, the paper could benefit from careful proof-reading of a native English speaker.
Author Response
Reviewer #1
1) The authors should provide Western blot analysis for AKT in their cell line models with/without application inhibitors and glucose starvation. All Western blot micrographs should contain information on the molecular weight of the detected bands. Also, histological images should contain a scale bar.
We thank the referee for asking us to include data regarding the status of the important protein AKT and to improve the presentation of our western blots. We now provide new data showing that the pattern of AKT phosphorylation does not follow that of the mTOR targets S6K, S6 and 4EBP. In short, QNZ and papaverine treatments result in variable AKT phosphorylation pattern. We show these data in Figure 4 and describe the data in the QNZ and papaverine selectively inhibit the mTOR pathway under glucose starvation section.
2) The xenografts should be assessed for a differential induction of apoptosis (e.g. cleaved caspase 3) and/or necrosis (conventional H&E stains). No statistics are provided for differences in tumor mass in Figure 5c, as well as for CD31 stains in Figure 5d. These should be added.
We have now added statistics, as requested.
Regarding the apoptosis analysis, while it is interesting, we argue that in-depth analysis of possible cell death in our tumor is beyond the scope of the present study. The available tissue was obtained at the end of the experiments were there are large differences in size between the control and treated tumors. Large tumors tend to develop necrosis independently of treatment.
3) Some figure labels and axes labels are very tine, especially in Figure 1. The authors should use the empty white space to enhance the readability of their figures by increasing the font size of the labels.
To make the figures more legible, we have increased the font size as requested.
4) The discussion section should be further expanded also pointing out the limitations of the study.
In the revised Discussion, we have added a “limitation of our study” section in which we focus on the limitations of our mechanistic explanation of why papaverine and QNZ enhance the activity of avastin in vivo.
5) References 30 and 35 are incorrectly spelled, and important reviews on 4EBP1 are not cited.
We have now reformatted and checked our references and added an appropriate reference to a review on 4EBP1
6) While the paper generally reads well, it contains numerous grammatical and orthographical errors (e.g. xerographs instead of xenografts; or incorrect use of verbs and nouns in divergent plural/singular form). Hence, the paper could benefit from careful proof-reading of a native English speaker.
Our manuscript has now gone through professional English editing.

Reviewer 2 Report
In this manuscript, Marciano R, Prasad M et al. conducted a high throughput screening to identify compounds that specifically target cancer cells under glucose starvation. Of 7000 agents, the list was reduced to 67 from which 31 agents show IC50 significantly more toxic under glucose starvation compared to normal conditions. Subsequently, out of this list 2 agents, papaverine and QNZ were selected for further validation using more than one cancer cell line. The authors found that papaverine and QNZ inhibit the mTOR pathway under glucose starvation. In vivo, the authors found that combination therapy with Avastin (VEGF inhibitor) exhibit a superior effect on tumor growth than each agent alone and suggest targeting mitochondrial activity as a potential therapeutic strategy in cancer.
Minor comments/corrections
Typos:
Lane 208 – probably the word “normal” is missing before “conditions”: …’could not be calculated under normal conditions.’
Fig 5A and C, in the graph of tumor volume please correct the word “vehicle”.
Fig 5B At the magnification that currently appears in the manuscript (IHC images) it is difficult to assess the lower expression of the markers selected. The authors can add a small window (inset) with an image in a higher magnification where the reader can appreciate the results summarized/quantified in the graph on the right side.
The authors keep mentioning along the manuscript the relevance of their identified hits in terms of mitochondrial targets under poor glucose conditions, and both agents have been previously reported to be mitochondria toxic. Since they are already showing the effect on the mTOR pathway, the link to mitochondria dysfunction is missing. I consider that the manuscript will gain more impact and significance if the authors are able to show the effect of the identified compounds on mitochondria function.
Author Response
Reviewer #2
Minor comments/corrections
Typos:
1) Lane 208 – probably the word “normal” is missing before “conditions”: …’could not be calculated under normal conditions.’
We have corrected these mistakes. In addition, the manuscript has undergone professional English editing.
2) Fig 5A and C, in the graph of tumor volume please correct the word “vehicle”.
We have corrected these mistakes.
3) Fig 5B At the magnification that currently appears in the manuscript (IHC images) it is difficult to assess the lower expression of the markers selected. The authors can add a small window (inset) with an image in a higher magnification where the reader can appreciate the results summarized/quantified in the graph on the right side.
We have added an inset to the images in Figure 5, as suggested.
4) The authors keep mentioning along the manuscript the relevance of their identified hits in terms of mitochondrial targets under poor glucose conditions, and both agents have been previously reported to be mitochondria toxic. Since they are already showing the effect on the mTOR pathway, the link to mitochondria dysfunction is missing. I consider that the manuscript will gain more impact and significance if the authors are able to show the effect of the identified compounds on mitochondria function.
We thank the reviewer for this suggestion, as inhibiting mitochondria is the best explanation we found for the activity of QNZ and papaverine. We measured mitochondrial membrane potential in cells treated with QNZ or papaverine with or without glucose starvation and found that both compounds inhibited mitochondrial membrane potential. These data are shown in new Figure 4B and are described in the QNZ and papaverine selectively inhibit the mTOR pathway under glucose starvation section.
Reviewer 3 Report
In their manuscript, Marciano et al performed a screen for drug compounds designed to selectively target glucose-deprived cancer cells. Two compounds found in this screen – papverine and QNZ – were shown to inhibit cell survival and promote apoptosis when assessed in vitro, and treatment of tumor-bearing mice with both QNZ and papaverine in combination with the VEGF inhibitor avastin / bevazicumab delayed tumor progression in vivo. Molecularly, the authors identified that under glucose starvation conditions the compounds selectively inhibit the mTOR pathway – a regulator of the glucose starvation response. Interestingly, about 50% of the effective compounds are mitochondrial toxins.
The study is interesting and provides insight into therapeutic potential to metabolically altered cancer cells. It is well planned and performed, however I have some concerns and suggestions to improve the paper – below.
Major comment:
1. Regarding the extended validation of the compounds on the two additional cell lines PANC1 and H1299. The reader wonders how these two cell lines were chosen, was it known that they undergo a metabolic change and rely less on glucose? Is there a biological basis for choosing these two cell lines? Were other cell lines also assayed, but without response? If so this is interesting and may contribute to revealing the mechanism.
2. Figure 3A – Please add the asterisks for the statistics on the graphs. Is the papaverine effect on PANC1 and H1299 significant?
3. Figure 3B – it would be nice to add a representative plot from the FACS showing the PI/Annexin V.
4. The authors performed combination treatment with papaverine and QNZ each with bevazicumab (avastin) and obtained synergy when testing in vivo. While these results are important, have the authors tested combination of papaverine and QMZ?
5. Line 276 –277 referring to the significant reduction in the levels of CD31 in all bevazicumab –treated tumors with reference to Fig. 5C and D. According to figure 5D – results are not significant, unless the authors forgot to include the asterisks on Fig 5D.
6. Line 293 – “…to uncover compounds acting towards (against) these…” the phrasing is somewhat confusing. Perhaps “that would intersect with”?
7. The supplementary figures are missing captions as well as descriptions/ figure legends.
Minor comments:
I encountered numerous typos while reading the manuscript, as well as some places where English language editing would be necessary. Some examples are:
1. Page 1 line 14 – in the Running Title should be deprived (not depraved).
2. “Normal Media” “normal conditions” these should be defined vis-à-vis glucose starvation.
3. Page 1, line 29: we found a high number.
4. Line 53 – there is an extra “)”
5. Line 154 – should be in plural: For experiments with more…”
6. Line 171 - viability dropped (not was dropping).
7. Line 217 – should be starvation (not staarvation)
8. Line 282 – xenograph – should be xenograft.
And many nore throughout the manuscript.

Author Response
We wish to thank the reviewer for the suggestions. We have amended the manuscript and figures accordingly. In particular, we added typical FACS plots and asterisks to denote statistical significance in Figure 3.
Point-by-point response to reviewer
1. Regarding the extended validation of the compounds on the two additional cell lines PANC1 and H1299. The reader wonders how these two cell lines were chosen, was it known that they undergo a metabolic change and rely less on glucose? Is there a biological basis for choosing these two cell lines? Were other cell lines also assayed, but without response? If so this is interesting and may contribute to revealing the mechanism.
The cell lines were chosen as “typical tumor cells” without considering the specific biological qualities of these cells. We have since tested the compounds using other tumor and non-tumor cell lines and found similar trends. Since other tested cell lines responded similarly, we did not follow up with more experiments and therefore show only the most validated and reproducible data.
2. Figure 3A – Please add the asterisks for the statistics on the graphs. Is the papaverine effect on PANC1 and H1299 significant?
The effect is significant and we added asterisk to denote this.
3. Figure 3B – it would be nice to add a representative plot from the FACS showing the PI/Annexin V.
We have now added typical FACS plots as requested.
4. The authors performed combination treatment with papaverine and QNZ each with bevazicumab (avastin) and obtained synergy when testing in vivo. While these results are important, have the authors tested combination of papaverine and QMZ?
We did not test the combination of papaverine and QNZ. We hypothesize that both compounds function through a similar mechanism, blocking mitochondrial complex I activity, and therefore do not expect the combination to be synergistic.
5. Line 276 –277 referring to the significant reduction in the levels of CD31 in all bevazicumab –treated tumors with reference to Fig. 5C and D. According to figure 5D – results are not significant, unless the authors forgot to include the asterisks on Fig 5D.
We added the missing asterisks.
6. Line 293 – “…to uncover compounds acting towards (against) these…” the phrasing is somewhat confusing. Perhaps “that would intersect with”?
To clarify our point, we changed the phrase “acting against” to “targeting”.
7. The supplementary figures are missing captions as well as descriptions/ figure legends.
The information is found in the Supplementary information section. Please let us know if we should add more information.
Minor comments:
I encountered numerous typos while reading the manuscript, as well as some places where English language editing would be necessary. Some examples are:
1. Page 1 line 14 – in the Running Title should be deprived (not depraved).
2. “Normal Media” “normal conditions” these should be defined vis-à-vis glucose starvation.
3. Page 1, line 29: we found a high number.
4. Line 53 – there is an extra “)”
5. Line 154 – should be in plural: For experiments with more…”
6. Line 171 - viability dropped (not was dropping).
7. Line 217 – should be starvation (not staarvation)
8. Line 282 – xenograph – should be xenograft.
We thank the reviewer for these comments. Our revised manuscript was correct by a professional English editor.